# Sub-Optimal Paternal Diet at the Time of Mating Disrupts Maternal Adaptations to Pregnancy in the Late Gestation Mouse

**DOI:** 10.3390/nu16121879

**Published:** 2024-06-14

**Authors:** Afsaneh Khoshkerdar, Nader Eid, Vipul Batra, Nichola Baker, Nadine Holmes, Sonal Henson, Fei Sang, Victoria Wright, Jane McLaren, Kevin Shakesheff, Kathryn J. Woad, Hannah L. Morgan, Adam J. Watkins

**Affiliations:** 1Lifespan and Population Health, School of Medicine, University of Nottingham, Nottingham NG7 2UH, UK; afsaneh.khoshkerdar@nottingham.ac.uk (A.K.); nader.eid@nottingham.ac.uk (N.E.); vipul.batra@nottingham.ac.uk (V.B.); nicky.baker@nottingham.ac.uk (N.B.); hannah.morgan@nottingham.ac.uk (H.L.M.); 2Deep Seq, School of Life Sciences, Queen’s Medical Centre, University of Nottingham, Nottingham NG7 2UH, UK; nadine.holmes1@nottingham.ac.uk (N.H.); sonal.henson1@nottingham.ac.uk (S.H.); fei.sang2@nottingham.ac.uk (F.S.); victoria.wright@nottingham.ac.uk (V.W.); 3Regenerative Medicine and Cellular Therapies, School of Pharmacy, University of Nottingham, Nottingham NG7 2UH, UK; jane.mclaren@nottingham.ac.uk (J.M.);; 4School of Veterinary Medicine and Science, University of Nottingham, Loughborough LE12 5RD, UK; katie.woad@nottingham.ac.uk

**Keywords:** paternal diet, maternal health, foetal programming, cardio-metabolic health

## Abstract

Pregnancy represents a stage during which maternal physiology and homeostatic regulation undergo dramatic change and adaptation. The fundamental purpose of these adaptations is to ensure the survival of her offspring through adequate nutrient provision and an environment that is tolerant to the semi-allogenic foetus. While poor maternal diet during pregnancy is associated with perturbed maternal adaptations during pregnancy, the influence of paternal diet on maternal well-being is less clearly defined. We fed C57BL/6 male mice either a control (CD), low protein diet (LPD), a high fat/sugar Western diet (WD) or the LPD or WD supplemented with methyl donors (MD-LPD and MD-WD, respectively) for a minimum of 8 weeks prior to mating with C57BL/6 females. Mated females were culled at day 17 of gestation for the analysis of maternal metabolic, gut, cardiac and bone health. Paternal diet had minimal influences on maternal serum and hepatic metabolite levels or gut microbiota diversity. However, analysis of the maternal hepatic transcriptome revealed distinct profiles of differential gene expression in response to the diet of the father. Paternal LPD and MD-LPD resulted in differential expression of genes associated with lipid metabolism, transcription, ubiquitin conjugation and immunity in dams, while paternal WD and MD-WD modified the expression of genes associated with ubiquitin conjugation and cardiac morphology. Finally, we observed changes in maternal femur length, volume of trabecular bone, trabecular connectivity, volume of the cortical medullar cavity and thickness of the cortical bone in response to the father’s diets. Our current study demonstrates that poor paternal diet at the time of mating can influence the patterns of maternal metabolism and gestation-associated adaptations to her physiology.

## 1. Introduction

During pregnancy, maternal physiology undergoes dramatic and dynamic adaptations, affecting almost every organ system including the cardiovascular, metabolic, microbiome and skeletal systems [1]. These maternal adaptations, which have the fundamental purpose of supporting the development of her offspring, initiate shortly after conception and continue throughout early neonatal life. Inappropriate maternal adaptation(s) can impair a range of processes including transport of nutrients via the placenta, utero-placental blood flow and the development and growth of the foetus. Such impairments can also result in the development of conditions such as gestational diabetes and pre-eclampsia, both major causes of maternal and foetal mortality [2,3]. Furthermore, gestational diabetes has been associated with foetal overgrowth, while pre-eclampsia results in growth restriction [4,5,6], both of which can impact the long-term cardio-metabolic health of the offspring. Therefore, appropriate maternal physiological responses during pregnancy, in both humans and animals, are critical for ensuring the well-being of both the mother and her offspring.

During a normal pregnancy, maternal metabolic status changes dramatically, which associates with weight gain, elevated fasting blood glucose levels, insulin resistance, glucose intolerance, low grade inflammation and altered metabolic hormone levels. During early pregnancy, typically referred to as the anabolic phase, there is an increase in maternal lipid accumulation driven by increased food intake and lipogenesis [7]. Underlying the promotion of fat storage is an increase in the levels of hormones such as progesterone, leptin, prolactin and cortisol in addition to significant adipose tissue hypertrophy [7,8,9]. In contrast, glucose homeostasis is maintained by increasing insulin secretion through hypertrophy and hyperplasia of pancreatic β-cells [10]. This is critical as glucose is essential for supporting foetal development. In women, during the third trimester, metabolism switches into a net catabolic phase, due mainly to a decrease in insulin sensitivity and an increase in the levels of oestrogen. The result of these changes are significant increases in serum total cholesterol, triglycerides and low-density lipoprotein cholesterol [11,12,13,14]. Additionally, studies have observed significant rise in the levels of high-density lipoprotein cholesterol [11,15]. The relative gestational hypertriglyceridemia is driven by both an increased accumulation and production of triglyceride-rich lipoproteins in combination with a decreased clearance by the liver. The rise in the levels of these maternal lipids is critical for providing the necessary additional energy required to support foetal growth and placental function. Furthermore, in late pregnancy, the influence of increased oestrogen levels and insulin resistance decreases the activity of adipose lipoprotein lipase and hepatic lipase activity [16], resulting in increased levels of circulating lipids [17] and an increase in fatty acid synthesis. Typically, by late pregnancy, levels of plasma cholesterol are approximately 30% higher than in non-pregnant women, while levels of triglyceride are approximately 50% higher [18].

Paralleling the changes to the maternal metabolic status are modulations of the cardiovascular system. Following conception, the uterine spiral arteries are remodelled, promoted by vacuolation of endothelial cells and swelling of the vascular smooth muscle in response to the immune cells within the decidua [19]. The array of uterine immune cells, including T regulatory cells, dendritic cells and macrophages, as well as the factors they secrete such as vascular endothelial growth factor (VEGF), placental growth factors (PlGF), transforming growth factor-beta (TGF-β) and granulocyte macrophage colony stimulating factor (CSF2) [20,21], enable trophoblast invasion and the initiation of spiral artery remodelling. This remodelling forms the basis of subsequent changes in the maternal cardiovascular system. By the eighth week of gestation in women, changes in cardiac output are detected which increases by up to 50% by weeks 16–20 [22]. Such changes are enabled through reductions in vascular peripheral resistance [23] and allow for an increased uteroplacental blood flow while maintaining maternal blood pressure.

Previously, we have shown that a paternal low protein diet (LPD), with or without the addition of 1-carbon methyl donors and carriers such as folate, methionine and vitamin B12, impacts on a range of paternal, embryonic, foetal, placental and adult offspring parameters [24,25,26,27,28]. We have also observed differential profiles of maternal uterine cytokines and gene expression of immune cell markers following mating with LPD fed males [25]. These observations indicate that the quality of a father’s diet at the time of mating impacts on both his developing offspring as well as the gestating mother. Paternally mediated changes in uterine vascular remodelling, embryo invasion and placental development and endocrine function could all influence maternal cardio-metabolic health and adaptations during pregnancy [29]. Such perturbations to maternal gestational physiology would also impact significantly on foetal well-being and long-term health. Therefore, the aim of this exploratory study was to explore the impact of paternal under (LPD) and over (a high fat/sugar Western diet; WD) nutrition on late gestation maternal cardio-metabolic status. Furthermore, we explore whether supplementation with a mix of methyl donors negates any detrimental influences of these poor-quality diets.

## 2. Materials and Methods

### 2.1. Animal Treatments and Tissue Collection

All experimental procedures were approved by the local ethics committee (AWERB) on the 06/10/2017 at the University of Nottingham. All procedures were conducted under the UK Home Office Animal (Scientific Procedures) Act 1986 Amendment Regulations 2012, which transposed Directive 2010/63/EU into UK law. Eight-week-old male C57BL/6J males and females (Harlan Ltd., Belton, Leicestershire, UK) were maintained within the Bio Support Unit at the University of Nottingham. Animals were housed in controlled 12/12 h light/dark conditions with a constant temperature (21 °C ± 3 °C) and access to food and water ad libitum. After a short period of acclimatization, males were allocated to one of five diets consisting of either a control diet (CD: 18% casein, 61% carbohydrate of which 21% are sugars, 10% fat), an isocaloric low protein diet (LPD: 9% casein, 69% carbohydrate of which 24% are sugars, 10% fat), the LPD supplemented with methyl donors and carriers (MD LPD; 5 g/kg diet choline chloride, 15 g/kg diet betaine, 7.5 g/kg diet methionine, 15 mg/kg diet folic acid, 1.5 mg/kg diet vitamin B12), a Western diet (WD: 19% casein, 44% carbohydrate of which 35% are sugars, 21% fat) or a Western diet supplemented with methyl donors and carriers (MD-WD; 5 g/kg diet choline chloride, 15 g/kg diet betaine, 7.5 g/kg diet methionine, 15 mg/kg diet folic acid, 1.5 mg/kg diet vitamin B12). All diets were commercially manufactured (Special Dietary Services Ltd., Witham, UK) and their full composition are detailed in Appendix A. Males (*n* = 8) were maintained on their respective diets for a minimum of 7 weeks prior to mating with virgin 8–12-week-old C57BL/6J females (*n* = 8). All females were mated with a separate male and maintained on standard chow (Rat and Mouse No.1 Maintenance chow diet, Special Dietary Services Ltd., Witham, UK). Pregnancy in females was confirmed by the presence of a vaginal plug (embryonic day 0.5; E0.5). Dams were culled via cervical dislocation at E17.5, and litter size was recorded. Samples of maternal liver and whole hearts were snap frozen in liquid nitrogen prior to storage at −80 °C. Maternal blood was collected via heart puncture, allowed to clot on ice and centrifuged at 10,000× *g*, 4 °C for 10 min prior to storage of the serum at −80 °C. Maternal faecal pellets were collected from the descending colon in nuclease-free tubes using sterile forceps and stored at −20 °C.

### 2.2. Assessment of Maternal Metabolic Status

Maternal liver cholesterol level was quantified using the Cholesterol Quantitation Kit (MAK043; Sigma-Aldrich, Gillingham, UK) in accordance with the manufacturer’s instructions. Briefly, approximately 10 mg of frozen liver tissue was ground to a powder using a pestle and mortar placed on dry ice. Samples were disrupted in 200 μL of a chloroform–isopropanol–IGEPAL (7:11:0.1; Sigma-Aldrich, Gillingham, UK) solution using a TissueLyser II (Qiagen, Manchester, UK). Samples were centrifuged at 13,000× *g* for 10 min to remove cellular debris. The organic phase was transferred to a new tube and dried under vacuum to remove excess chloroform. Samples were reconstituted in an assay buffer, prior to incubation with a cholesterol probe and a cholesterol esterase enzyme mix, in accordance with the manufacturer’s instructions. Samples were incubated at room temperature for 60 min and analysed in duplicate at 585 nm using an iMark Microplate Reader (Bio-Rad, Watford, UK).

Maternal liver free fatty acid levels were quantified using the Free Fatty Acid Quantitation Kit (MAK044; Sigma-Aldrich, Gillingham, UK) in accordance with the manufacturer’s instructions. Briefly, approximately 10 mg of frozen liver tissue was ground to a powder using a pestle and mortar placed on dry ice prior to disruption in 200 μL of a 1% (*w*/*v*) Triton X-100 in chloroform (Sigma-Aldrich, Gillingham, UK) solution using a TissueLyser II (Qiagen, Manchester, UK). The sample was centrifuged at 13,000× *g* for 10 min. The organic phase was removed and dried under vacuum before being reconstituted in assay buffer, prepared in accordance with the manufacturer’s instructions. Samples were analysed in duplicate at 570 nm using an iMark Microplate Reader (Bio-Rad, Watford, UK).

Maternal liver triglyceride levels were measured using the Triglyceride Quantification Kit (MAK266; (Sigma-Aldrich, Gillingham, UK) in accordance with the manufacturer’s instructions. Briefly, approximately 100 mg of frozen liver tissue was ground to a powder using a pestle and mortar placed on dry ice prior to disruption in 1 mL of 5% Nonidet P40 (Sigma-Aldrich, Gillingham, UK) using a TissueLyser II (Qiagen, Manchester, UK). The sample was centrifuged at 13,000× *g* for 10 min prior to analysis of the supernatant in accordance with the manufacturer’s instructions. Samples were analysed in duplicate at 595 nm using an iMark Microplate Reader (Bio-Rad, Watford, UK).

Maternal Serum glucose levels were determined using a glucose colorimetric detection kit (EIAGLUC; Thermo Fisher Scientific, Loughborough, UK) in accordance with the manufacturer’s instructions. Samples were measured in duplicate at 595 nm using an iMark Microplate Reader (Bio-Rad, Watford, UK). Serum insulin levels were determined using a rat/mouse insulin ELISA kit (EZRMI-13K; EMD Millipore Corporation, Burlington, MA, USA) in accordance with the manufacturer’s instructions. Samples were measured in duplicate at 450 nm using an iMark Microplate Reader (Bio-Rad, Watford, UK).

### 2.3. Maternal Gut Microbiota Sequencing

DNA was isolated from maternal faecal pellets using the QIAamp DNA stool mini kit (Qiagen, Manchester, UK) in accordance with the manufacturer’s instructions. Sequencing was conducted as described previously [28]. Briefly, sequencing was conducted on the V3-V4 region of the 16S rRNA gene in accordance with Illumina 16S Metagenomic Sequencing Library Preparation protocol. The 16S rRNA amplicons were generated using forward 5′ (TCGTCGGCAGCGTCAGATGTGTATAAGAGACAGCCTACGGGNGGCWGCAG) and Reverse 5′ (GTCTCGTGGGCTCGGAGATGTGTATAAGAGACAGGACTACHVGGGTATCTAATCC) primers, flanked by Illumina adapter-overhang sequences. Illumina dual index barcodes (Illumina XT Index Kit v2, Set A: FC-131-2001; Illumina, Cambridge, UK) were attached to each amplicon. PCR clean-up was conducted using AMpure XP beads (Beckman; A63882; High Wycombe, UK). Library fragment-length distributions were analysed using the Agilent TapeStation 4200 and the Agilent D1000 ScreenTape Assay (Agilent; 5067-5582 and 5067-5583; Stockport, UK). Libraries were pooled in equimolar amounts, and the pool was size-selected using the Blue Pippin (Sage Science; Beverly, MA, USA) and a 1.5% Pippin Gel Cassette (Sage Science; BDF2010; Beverly, MA, USA). Sequencing was performed by Deep Seq at the University of Nottingham on an Illumina MiSeq using a MiSeq Reagent Kit v3 (600 cycle) (Illumina; MS-102-3003; Illumina, Cambridge, UK) to generate 300 bp paired-end reads. Raw reads were processed by the Qiime2 pipeline and trimmed. Greengenes version 13.8 was used in the classification [30].

### 2.4. Maternal Liver RNA-Seq

Total RNA was isolated from samples of maternal liver using the RNeasy Mini plus kit (Qiagen, Manchester, UK), in accordance with the manufacturer’s instructions. RNA quantity and quality were initially assessed by Nanodrop. Stranded RNA-seq libraries were prepared from 500 ng of total RNA per sample, using the NEBNext Ultra II Directional RNA Library Preparation Kit for Illumina (NEB; E7760) and NEBNext Multiplex Oligos for Illumina (96 Unique Dual Index Pairs) (NEBNext; E6440). Prior to library preparation, total RNA was treated with QIAseq FastSelect -rRNA HMR (Cat: 334222, Qiagen, Manchester, UK) to prevent any rRNA present from being converted into the sequencing library. Libraries were quantified using the Qubit Fluorometer and the Qubit dsDNA HS Kit (ThermoFisher Scientific; Q32854; Loughborough, UK). Library fragment-length distributions were analysed using the Agilent 4200 TapeStation and the Agilent High Sensitivity D1000 ScreenTape Assay (Agilent; 5067-5584 and 5067-5585; Stockport, UK). Libraries were pooled in equimolar amounts and final library quantification was performed using the KAPA Library Quantification Kit for Illumina (Roche; KK4824; Welwyn Garden City, UK). The library pool was sequenced on the Illlumina NextSeq500 over three NextSeq500 High Output 150 cycle kits (Illumina; 20024907; Cambridge, UK) to generate over 40 million pairs of 75 bp paired-end reads per sample. Raw reads were trimmed of Illumina adapters and low quality (Q < 20) nucleotides using TrimGalore (v 0.6.7). Reads shorter than 15 bp were discarded. Trimmed reads were aligned to the Mus musculus reference genome GRCm39 using HISAT2 (v 2.2.1). StringTie (v 2.2.1) was used to assemble genes and calculate gene abundance. Differential expression analysis was performed using DESeq2 (within version 3.0 of Bioconductor). Liver RNA-seq data have been submitted to the Gene Expression Omnibus (GEO) at NCBI under accession number: GSE265783.

### 2.5. Maternal Femur µ-CT Analysis

Maternal femurs were dissected free from muscle prior to fixation in 4% neutral buffered formalin (Sigma, UK) at room temperature overnight and subsequent storage in 70% ethanol prior to analysis. Whole femurs were scanned using a Skyscan 1174 µ-CT scanner (Bruker, Belgium). All scans were taken at 50 kVa and 800 µA with a 0.5 mm aluminium filter, 3600 ms exposure time, 180° tomographic rotation and a Voxel resolution of 17.84 µm^2^. Individual two-dimensional cross-sectional images were reconstructed using Bruker NRecon software (version 1.7.4.6). Stacks of images for each individual femur were imported into BoneJ [31], and the total number of sections was defined. Identification of the trabecular and cortical regions for analysis was defined based on the first appearance of a bridging connection of the low-density growth plate chondrocyte seam. Using this as a standard anatomical set point, an offset of 3% of the total bone length towards the femoral head from the reference growth plate was used to define the start of the trabecular region of interest, and a series of trabecular sections representing 5% of the total bone length was analysed. For the analysis of the cortical bone, the mid bone position was defined based on the total length of the bone, and an offset of 2.5% of the total bone length towards the femoral head was used to define the start of the cortical region. A series of cortical sections representing 5% of the total bone length was analysed. Using BoneJ, measurements of trabecular and cortical bone volume (Bv), total volume (Tv), bone volume fraction (Bv/Tv), trabecular thickness (Tb.Th), medullary cavity volume (Mv) degree of anisotropy (Da), connectivity density (Con.D), maximum moment of inertia (Imax) and minimum moment of inertia (Imin) were defined. In addition, cortical bone volume (Cb-V), cross section area (Cb-Cx) and thickness (Cb-Th) and the volume of the medullary cavity (Cb-Mv) were also defined.

### 2.6. Maternal Cardiac Gene Expression

Total RNA was extracted from frozen maternal heart tissue using the RNeasy Plus Mini Kit (Qiagen, Manchester, UK). Whole hearts were powdered using a pestle and mortar over dry ice. No more than 25 mg of powdered tissue was disrupted in the kit’s lysis buffer using the TissueLyser II (Qiagen, Manchester, UK) prior to RNA isolation in accordance with the manufacturer’s instructions. cDNA synthesis was performed using the TaqMan™ Reverse Transcription Reagents kit (N8080234; ThermoFisher Scientific, Loughborough, UK) according to manufacturer’s protocol with 1 μg input of RNA. Real-time quantitative PCR (RT-qPCR) was performed as previously outlined [27]. Briefly, reactions consisted of 5 ng cDNA and 175 nM forward and reverse primers (Eurofins Genomics, Ebersberg, Germany) with 1× Precision SYBR Green Mastermix (PrimerDesign, Southampton, UK), and RT-qPCR was performed using an Applied Biosystems 7500 Fast system. Gene expression was analysed using the delta–delta Ct method relative to CD expression. The GeNorm method used to normalise gene expression, as previously described [32] to two reference genes phosphoglycerate kinase 1 (Pgk1) and tubulin, alpha 1a (Tuba1a). All primer sequences are detailed in Appendix A.

### 2.7. Statistical Analyses

All maternal (*n* = 8) data were analysed using GraphPad Prism (version 10) or SPSS (version 28). Data were assessed initially for normality (Shapiro–Wilk and Kolmogorov–Smirnov tests) prior to analysis using one-way ANOVA for normally distributed data or a Kruskal–Wallis test for non-normally distributed data with an appropriate post hoc test for correction of multiple group comparisons. Where appropriate, data were corrected for multiple comparisons using the Benjamini–Hochberg False Discovery Rate (FDR) method. Correlations between parameters were conducted using Pearson correlation. Significance was taken at *p* < 0.05.

## 3. Results

### 3.1. Maternal Physiology and Metabolic Status

On the day immediately after mating (E0.5), there were no significant differences in mean body weight between any of the female groups. Similarly, there were no differences in mean maternal body weight when the females were culled at gestational day 17.5 (E17.5; Figure 1A), in the % weight gain from E0.5 to E17.5 (Figure 1B) or in the raw weight of the maternal heart (Figure 1C) or liver (Figure 1D). Additionally, there were no differences in mean litter size between groups (Figure 1E). While there were minimal differences between the groups in relation to mean body/organ weights, we identified group-specific correlative traits. In all groups, we identified significant positive correlations between maternal weight at day E17 and total weight gained over pregnancy. In females mated with CD fed males, we also observed a positive association between maternal E17.5 weight and the weight of the heart, liver and kidneys (Figure 1F, *p* < 0.03). However, in females mated with LPD and MD-LPD fed males, only a positive association between maternal weight and kidney weight existed (Figure 1G,H), while in in females mated with WD and MD-WD fed males, the association was between maternal weight and liver weight (Figure 1I,J). In females mated with CD fed males, we observed significant negative correlations between weight gained per foetus, maternal weight gain and litter size (Figure 1F, *p* < 0.03). However, such negative associations between increases in litter size (and hence maternal weight gain) and weight per foetus were not observed in any of the other groups (Figure 1G–J). Analysis of maternal hepatic and serum metabolite levels revealed no difference in mean hepatic triglyceride (Figure 1K), free fatty acids (Figure 1L) and cholesterol (Figure 1M), or in serum insulin (Figure 1N) or glucose (Figure 1O).

### 3.2. Impact of Paternal Diet on Late Gestation Maternal Gut Microbiota

To investigate whether paternal diet influenced late gestation maternal gut bacterial diversity, we sequenced the hypervariable V3–V4 region of the bacterial 16S rRNA gene (see Appendix A). We observed no difference in bacterial diversity between groups (Figure 2A,B). Principle component analysis (PCA) showed all groups clustered similarly (Figure 2C). Analysis of bacterial profiles at the phylum (Figure 2D) and family (Appendix A) levels showed no significant differences between treatment groups. Furthermore, the assessment of the relationship between the abundance of Bacteroidetes and Firmicutes showed no differences between groups (Figure 2E). However, to understand whether poor paternal diet may influence the broader maternal bacterial profiles, we correlated each bacterial component at the family level with each other. All groups showed a significant positive correlation between the abundance of Actinobacteria and Bifidobacteriaceae (*p* < 0.05; Figure 2F–J), as well as a significant negative correlation between the abundance of Firmicutes and Bacteroidetes (*p* < 0.05; Figure 2F–J). In females mated with CD fed males, we observed a significant positive correlation between Tenericutes and Proteobacteria (r = 0.99, *p* < 0.0001; Figure 2F) which was not present in any of the other groups. In females mated with LPD fed males, we observed a significant positive association between the abundance of Proteobacteria and Actinobacteria (r = 0.87, *p* = 0.024; Figure 2G), an association also seen in females mated to WD fed males (r = 0.95, *p* = 0.004; Figure 2I) as well as a negative association between unclassified bacteria, termed ‘Others’ and Firmicutes (r = −0.82, *p* = 0.046). In females mated with MD-LPD fed males, we observed a positive correlation between ‘Others’ and Protobacteria (r = 0.92, *p* = 0.001), as well as significant associations between the abundance of Bifidobacteriaceae with ‘Others’ (r = 0.90, *p* = 0.014) and with Tenericutes (r = 0.091; *p* = 0.013) (Figure 2H). In females mated with MD-LPD fed males, we also observed significant positive correlations between Deferribacteres and Firmicutes (r = 0.89, *p* = 0.017; Figure 2H) and a negative association between Deferribacteres and Bacteroidetes (r = −0.95, *p* = 0.004; Figure 2H), associations which were not present in the other groups. In females mated with WD fed males, we observed a significant positive correlation between Proteobacteria and Bifidobacteriaceae (r = 0.91, *p* = 0.01; Figure 2I) and a negative association between Proteobacteria and Firmicutes (r = −0.87, *p* = 0.024; Figure 2J). Finally, in females mated with MD-WD fed males, we observed a positive association between ‘Others’ and Protobacteria (r = 0.92, *p* = 0.008; Figure 2J).

### 3.3. Comparison of Maternal Late Gestation Liver Transcriptome

To explore further the potential influences of paternal diet on maternal metabolic status, we conducted RNA-Seq on samples of maternal liver (RNA-Seq data are currently being deposited in the Gene Expression Omnibus). Using a *p*-value of < 0.05 and no log2 fold-change cut-off, we detected 743, 934, 540 and 508 differentially expressed genes between females mated with CD and LPD, MD-LPD, WD and MD-WD males, respectively (Figure 3A–D). We conducted subsequent pathway and ontology analysis using differentially expressed genes with a *p*-value of < 0.05 and no log2 fold-change cut-off in order to obtain a broader understanding of potentially subtle changes in maternal metabolic status and the interplay between them. Interestingly, we observed minimal overlap in the number of shared up- and down-regulated genes between each group (Figure 3E,F). Between females mated with LPD and MD-LPD males, there were only 33 uniquely shared down-regulated genes (Figure 3E). Similarly, between females mated with WD and MD-WD males, we identified only 15 unique genes in common between the two groups. When comparing the differentially expressed down-regulated gene profiles across all four experimental diet groups, 17 genes were common to all four diets (Figure 3E). A similar pattern was observed for common up-regulated genes. Here, 56 and 28 genes were common between females mated with LPD and MD-LPD fed males and between females mated with WD and MD-WD fed males, respectively (Figure 3F). Only one single gene, Src-like-adapter 2 (Sla2), was upregulated in all four groups (Figure 3F). In total, 23 genes were identified as being significantly differentially expressed in all groups (Figure 3G). Cluster analysis of these common genes indicated that females mated with WD and MD-WD males clustered together, while females mated with LPD fed males showed the most variability in levels of expression. These data indicate that each of the paternal diets has a largely unique impact on the transcriptomic profile in the maternal liver.

Pathway and gene ontology analysis of the differentially expressed maternal genes identified significant (Padj < 0.01) upregulation of genes associated with protein transport (GO:0015031), lipid metabolism (GO:0006629) and endosome organisation (GO:0007032) in females mated with LPD fed males when compared to female mated with CD fed males (Figure 3H). In contrast, pathways associated with mRNA splicing (GO:0008380) and processing (GO:0006397) were significantly (Padj < 0.01) down regulated. In females mated with MD-LPD fed males, genes associated with protein transport were also upregulated (Figure 3I), regulation of transcription and ubiquitin-dependent protein catabolic process (GO:0006511) and the respiratory chain (GO:0042773) were also upregulated (Padj < 0.01). In contrast, multiple genes associated with innate immunity, chemotaxis and stress responses were down regulated when compared to females mated with CD fed males. In females mated with WD fed males, only ubiquitin-dependent protein catabolic (GO:0006511) and transcription (KW-0804) processes (Figure 3J) were significantly up- and down-regulated, respectively (Padj < 0.05). Finally, in females mated with MD-WD fed males, we observed up-regulation of genes involved in ubiquitin-dependent protein catabolic process (GO:0006511) and protein transport (GO:0015031), while genes associated with heart development (GO:0007507) and cell–cell adhesion (GO:0098609) were decreased (Figure 3K) when compared to females mated with CD fed males (Padj < 0.001).

### 3.4. Maternal Late Gestation Cardiac Gene Expression

To determine if maternal cardiac health may be altered in response to poor paternal diet, we assessed the expression of genes central in the regulation of cardiac function. We observed no difference in the Ct values of the reference genes phosphoglycerate kinase 1 (Pgk1) or tubulin, alpha 1A (Tuba1a) (Figure 4A,B). Additionally, we observed no difference in the relative expression of angiotensin converting enzyme 2 (Ace2, Figure 4C), adrenergic receptor, beta 1 (Adrb1, Figure 4D), angiotensin II receptor, type 1a (Agtr1a, Figure 4E) or the ATPase, Ca2+ transporting, plasma membrane 1 (Atp2b1, Figure 4F) between groups.

### 3.5. Maternal Late Gestation Femur Trabecular Bone Morphology

Finally, to determine whether paternal diet at the time of conception could also affect other maternal systems, we analysed maternal femur morphology by µCT. Initially, we observed that femurs from females mated with MD-LPD fed males were significantly longer than femurs from CD mated females (*p* = 0.023; Figure 5A). However, there were no differences in mean femur volume between any of the groups (Figure 5B). Morphological analysis of the trabecular bone (Figure 5C) identified significant differences in total trabecular volume (Tv) between females mated with MD-LPD fed males and females mated with MD-WD fed males (*p* = 0.047; Figure 5D). However, there were no differences in trabecular bone volume (BV; Figure 5E), in Bv:Tv ratio (Figure 5F) or in trabecular thickness (Tb:Th; Figure 5G). Finally, analysis of the number of connected trabeculae in the image (ConD) revealed a significant reduction in females mated with MD-LPD fed males when compared to female mated to LPD fed males (*p* = 0.049; Figure 5H).

### 3.6. Maternal Late Gestation Femur Cortical Bone Morphology

Analysis of cortical bone morphology (Figure 6A) indicated all females had a similar volume of cortical bone (C-Bv; Figure 6B). However, females mated with males fed a MD-WD had a reduced volume of the cortical medullary cavity when compared to female mated with CD. LPD and WD fed males (Cb-Mv, *p* < 0.05; Figure 6D). Similarly, females mated with males fed a MD-WD displayed an increase in mean cortical bone thickness (Cb-Th) when compared to all other groups (*p* < 0.05; Figure 6E). However, there were no differences in the mean cortical bone cross-section areas between groups (Figure 6F). Finally, analysis of the second moment of area around major (Imax) and minor (Imin) axis showed no differences between groups (Figure 6G,H).

## 4. Discussion

The impacts of poor maternal diet in pregnancy on the health of her offspring, as well as her own health, are well-established [33]. However, the influence that a father has on this fundamental period is less well-defined. In the current study, we explored the impact of paternal nutrition on maternal cardio-metabolic and bone health in late gestation. We observed subtle changes in patterns of association between maternal weight gain, organ weight and litter size despite no change in metabolic status. In contrast, we observed differential hepatic expression of genes associated with protein transport, lipid metabolism, transcription, innate immunity, chemotaxis and ubiquitin process in females mated with males fed sub-optimal diets. Finally, we observed differential remodelling of cortical and trabecular regions of the maternal femur in response to paternal diet. These data highlight the broad influences that paternal nutrition can have on maternal health during pregnancy, irrespective of her own dietary status.

We observed minimal differences in maternal weight gain or organ sizing during pregnancy between groups. However, we observed subtle differences in the way maternal weight gain and organ weight interacted. Additionally, we observed a negative association between litter size and the amount of weight gained by the dam in females mated with CD fed males. This observation suggests that as litter size increases, so does the metabolic demand on the dam, and the female’s resources are utilised to a greater extent, resulting in a diminishing body weight. Interestingly, only females mated with MD-WD fed males also showed this association. In late gestation, females have entered a catabolic state of metabolism, characterised by an increased breakdown of adipose deposits and liberation of non-esterified fatty acids and glycerol. The non-esterified fatty acids can be oxidised to acetyl-CoA, while glycerol can be utilised in glucose synthesis [33]. While the changes in patterns of gestational body size might suggest differential nutritional partitioning between groups, we observed no significant differences in circulating or hepatic metabolites.

To establish how paternal diet might influence maternal metabolic status in more detail, we next analysed the maternal microbiota. The significance of the gut microbiota in the metabolism and synthesis of vitamins [34], regulation of lipid metabolism [35], inflammatory [36] and metabolic diseases [37] as well as cardiovascular disease [38] is now accepted. In this study, we sampled from the lower region (ileum and post ileocecal) of the gut which is predominated by coliforms and anaerobic species including Bacteroides, Bifidobateria, Clostridia and Lactobacilli [39]. In all females, a significant positive correlation between the abundance of Actinobacteria and Bifidobacteriaceae was observed. Actinobacteria are a diverse phylum of Gram-positive bacteria which includes the anaerobe families (Bifidobacteria, Propionibacteria and Corynebacteria) and an aerobe family (Streptomyces) [40]. Levels of Bifidobacteria have been inversely linked with BMI and insulin levels in women [41]. In contrast, reduced levels of Bifidobacteria have been linked to enhanced gut permeability [42] and immune system activation [42]. Therefore, the lack of any difference in the abundance of these bacterial groups is in line with no significant differences in the central metabolic status. Similarly, all females showed a negative association between Firmicutes and Bacteroidetes. The Firmicutes–Bacteroidetes ratio is widely cited as a marker of obesity with a higher ratio seen in obese mice and women when compared to females of a healthy weight [43]. However, as all our females displayed the same overall weight gain and body morphometry then a change in the Firmicutes–Bacteroidetes ratio might not be anticipated. However, some subtle differences in the associations between different bacterial groups were observed. Females mated with LPD fed males showed a positive association between the abundance of Actinobacteria and Protobacteria, while females mated with WD fed males showed a positive association between the abundance of Protobacteria and Bifidobacteriaceae. As the gut microbiota has a wide range of influences on physiology, subtle shifts in patterns of relative abundance could still have influences on maternal health.

To define the wider effects of poor paternal diet on maternal health in pregnancy, we conducted RNA-Seq analysis on the maternal liver. Females mated with LPD and MD-LPD fed males displayed the largest number of differentially expressed genes, 743 and 934 genes, respectively. However, only 89 genes were common to both groups. Here, pathway analysis identified changes in lipid metabolism, mRNA processing, protein transport and immunity. Increases in the expression of genes such as phosphatidate cytidylyltransferase 2 (*Cds2*), adiponectin receptors 1 and 2 (*Adipor 1/2*), long-chain fatty acid transport protein 2 (*Slc27a2*), sterol O-acyltransferase 1 (*Soat1*), cardiolipin synthase 1 (*Crls1*) and several genes involved in acyl-CoA metabolism (*Acaca*, *Acsf2*, *Acss3*, *Acadl*, *Acads*) in females mated to LPD fed males suggest differential patterns of lipid uptake and metabolism within the mitochondria. Beta-oxidation is a source of significant metabolic energy during periods of high energy demand [44]. Impairments in β-oxidation have been linked to the accumulation of hepatic lipid-species, mitochondrial dysfunction and the development of metabolic diseases such as non-alcoholic fatty liver disease [45]. Therefore, a more detailed lipidomic profiling, in combination with mitochondrial function, metabolism and lipid composition, in our females is warranted.

In females mated with MD-LPD males, we observed differential regulation of genes associated with transcription and immunity. The immune system modulates throughout pregnancy, shifting from a pro-inflammatory state in early gestation to a more anti-inflammatory state in later gestation [46]. This concept is supported by observations that the severity of autoimmune disorders appears diminished during pregnancy but then returns post-partum [47]. Other conditions such as multiple sclerosis also show decreased progression during pregnancy [48]. We observed a decreased expression of multiple complement genes (*C8a*, *C8b*, *C9*), histocompatibility genes (*H2-D1*, *H2-Ab1*, *H2-Eb1*), antigen presentation (*Fcgr1*, *Tapbpl*), and inflammatory mediators (*Card9*, *Cx3cr1*, *Mefv*, *Tlr12*). Changes in maternal immunological status are directly linked to hormonal levels. Progesterone has been shown to inhibit the proliferation and cytokine secretion of CD8+ T cells [49], while in T-helper cells, progesterone promotes the expression of LIF [50], both of which would support the maintenance of pregnancy. Whether the potential immunological changes observed in our current study are due to paternal influences on maternal hormonal status in pregnancy are currently unknown as maternal endocrine homeostasis is still to be determined in this model. It is also unclear whether any potential immunological perturbations in our females might put the dams, and their offspring, at greater risk of infection. Finally, we are unable to ascertain whether the reduction in maternal immunological status is a paternally mediated adaptive mechanism to increase maternal tolerance of his offspring, thus enhancing their survival.

In females mated with WD and MD-WD fed males, we observed 540 and 508 differentially expressed genes, respectively, with only 43 genes shared exclusively between them. Gene ontology and pathway analysis identified few significant changes in females mated with WD fed males. Genes associated with the ubiquitin pathway were up-regulated while genes involved in transcription were down-regulated. In females mated with MD-WD fed males, we also observed an up-regulation of ubiquitin pathway genes suggesting an increase in protein turnover in both groups. In contrast, a down regulation of cardiac and cardiovascular genes was also observed in these females. Closer analysis of these genes identified regulators of transcription (*Kdm6b*), vessel formation and angiogenesis (*Apln*, *Tgfb2*), cell and muscle structure (*Flrt3*, *Myh10*, *Ttn*). Whether any potential disruptions in hepatic immune status and vascular function in our females are also reflective of wider cardiovascular and immune impairments remains to be determined. However, such impairments in pregnancy have been associated with gestational conditions such as preeclampsia [51]. However, we observed no differences in central cardiac regulatory gene expression between groups.

Our final observation was that poor paternal diet influenced maternal femur structure in late gestation. We observed modest changes in femur length, volume of trabecular bone (Tv), trabecular connectivity (ConD), volume of the cortical medullar cavity (Cb-Mv) and thickness of the cortical bone (Cb-Th) between groups. Similar to many of the other maternal systems, the skeleton undergoes dramatic adaptation throughout pregnancy so as to provide sufficient calcium and other minerals to support foetal development [52]. Underlying the modulation of maternal bone physiology is parathyroid hormone-related protein (PTHrP). During pregnancy, PTHrP production by the placenta and breasts increases due to the action of oestradiol, placental lactogen and prolactin [52]. During early pregnancy, an increase in intestinal calcium absorption is sufficient to meet the foetal demands [53]; however, in later gestation and during lactation, the decline in oestradiol levels in combination with PTHrP results in greater maternal bone resorption [54,55]. During lactation, there is a continued (3–10%) loss in bone mineral density in women which occurs predominantly in the trabecular bone (i.e., lumbar spine) and some cortical regions such as the hip [56]. However, much of this is reversed by 6–12 months after breast feeding stops [57,58]. While studies indicate that maternal bone physiology can recover post-lactation, the short- and long-term effects of inappropriate bone remodelling during pregnancy are unknown.

## 5. Conclusions

While our study indicates that poor paternal diet at the time of conception can influence maternal physiological and metabolic status in late gestation, the underlying mechanisms remain unanswered. One central mechanism is a change in sperm epigenetic status. Previously, we have shown that sperm from LPD fed males display genome-wide DNA hypomethylation and the expression of several epigenetic regulators within the testis of LPD fed males was perturbed [25]. Other studies have linked paternal diet [59], ageing [60], and environmental pollutants [61] to perturbed patterns of offspring health and placental development. As many aspects of maternal health in pregnancy are regulated via placental function, metabolism and the range of endocrine factors it produces [62], paternal epigenetic modulation of the placenta provides one link between paternal diet and maternal health.

Our study widens our understanding of the range of influences that poor paternal diet has on the environment in which his offspring development. There are several limitations within our findings. First, the use of a methyl-donor-supplemented control diet (MD-CD) would have provided additional insight into the role that the methyl donors on their own might play. However, comparing the size of effect between the respective diets with (MD-LPD, MD-WD) and without (LPD, WD) methyl-donor supplementation suggests that they are having a minimal impact on maternal health. Similarly, our WD is deficient in methionine when compared to the other diets. However, as minimal differences were seen in the effects of WD vs. MD-WD, we do not believe this to be a significant factor for maternal well-being. Also, as already discussed, our understanding of the maternal metabolic changes is currently limited. Future studies will adopt a wider analysis of maternal serum lipid and hormone profiles, as well as microbiota-associated metabolites such as butyrate and other short-chain fatty acids. A final limitation of this study was that we were unable to perform a full characterisation of maternal cardiovascular health. Analysis of blood pressure, vascular reactivity or cardiac morphology would have provided a more insight into how paternal diet affects maternal cardiovascular health in pregnancy.

## Figures and Tables

**Figure 1 nutrients-16-01879-f001:**
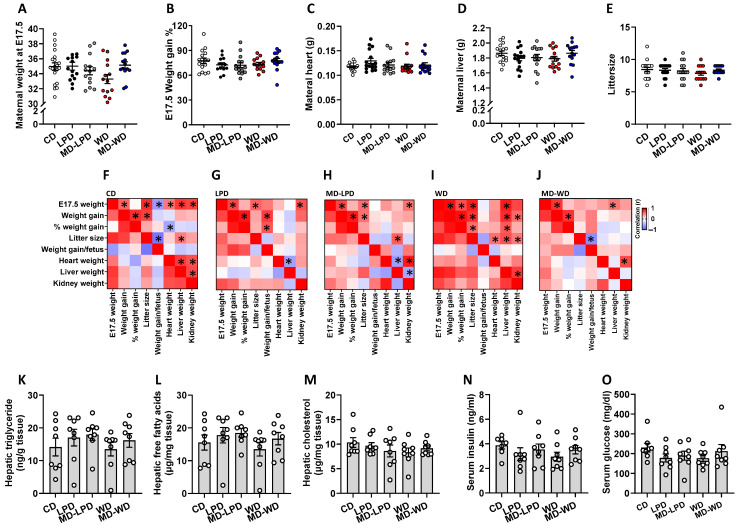
Impact of paternal diet on late gestation maternal physiology and metabolic status. Maternal weight immediately following mating on embryonic day (E)0.5 (**A**) and at the time of cull on E17.5 (**B**) in females mated to males fed either a control diet (CD), low protein diet (LPD), methyl donor-supplemented LPD (MD-LPD), Western diet (WD) or methyl donor-supplemented WD (MD-WD). Maternal heart (**C**) and liver (**D**) weight and litter size (**E**) at E17.5. Correlations between maternal weight at E17, gestational weight gain (E0.5–17.5), weight gain as a % of body weight, litter size, weight gain/foetus, heart, liver, and kidney weight in females mated to CD (**F**), LPD (**G**), MD-LPD (**H**), WD (**I**) or MD-WD (**J**) fed males (red boxes denote positive correlations, blue denotes a negative correlation, boxes with an asterisk denote statistically significant correlations). Late gestation maternal hepatic triglyceride (**K**), free fatty acids (**L**) and cholesterol (**M**) and serum insulin (**N**) and glucose (**O**) levels. *n* = 8 females per treatment group, each mated with a separate male. Data are expressed as mean ± SEM (**A**–**E**,**K**–**O**) or Spearman’s correlation (**F**–**J**). Statistical differences were determined using a one-way ANOVA or Kruskal–Wallis test with post hoc correction (**A**–**E**,**K**–**O**), or by Spearman’s correlation (**F**–**J**). Statistical significance was taken when *p* < 0.05.

**Figure 2 nutrients-16-01879-f002:**
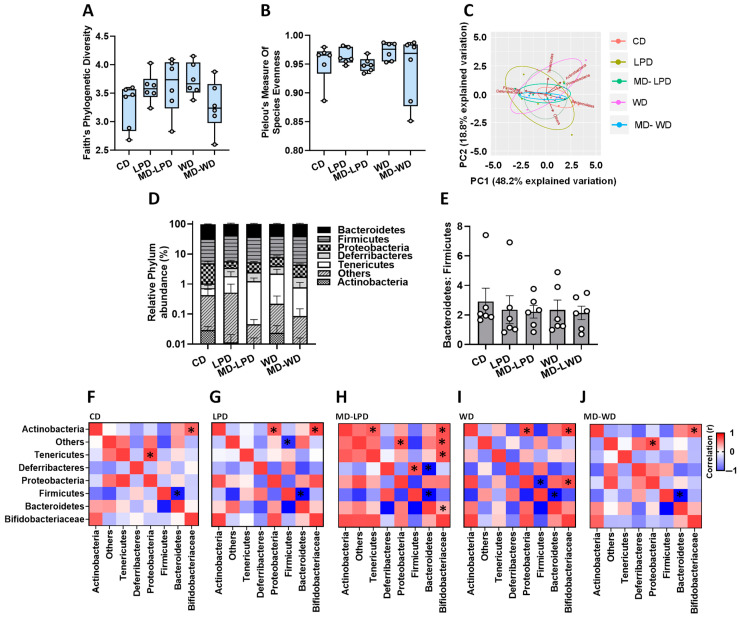
Impact of paternal diet on late gestation maternal gut microbiota. Assessment of maternal faecal bacterial profiles using Faith’s phylogenetic diversity (**A**) and Pielou’s measure of species evenness (**B**) in females mated to males fed either a control diet (CD), low protein diet (LPD), methyl donor-supplemented LPD (MD-LPD), Western diet (WD) or methyl donor-supplemented WD (MD-WD). Principal component analysis (PCA) based on bacterial abundance at the phylum level (**C**) and relative abundance (**D**). Ratio of gut Bacteroidetes to Firmicutes (**E**). Correlations between bacterial populations in females mated to males fed either CD (**F**), LPD (**G**), MD-LPD (**H**), WD (**I**) or MD-WD (**J**) (boxes with an asterisk denote statistically significant correlations). *n* = 6 females per treatment group, each mated with a separate male. Data are expressed as mean ± minimum to maximum value (**A**,**B**), mean ± SEM (**D**,**E**) or Spearman’s correlation (**F**–**J**). Statistical differences were determined using a one-way ANOVA or Kruskal–Wallis test with post hoc correction (**A**,**B**,**D**,**E**), or by Spearman’s correlation (**F**–**J**). Statistical significance was taken when *p* < 0.05.

**Figure 3 nutrients-16-01879-f003:**
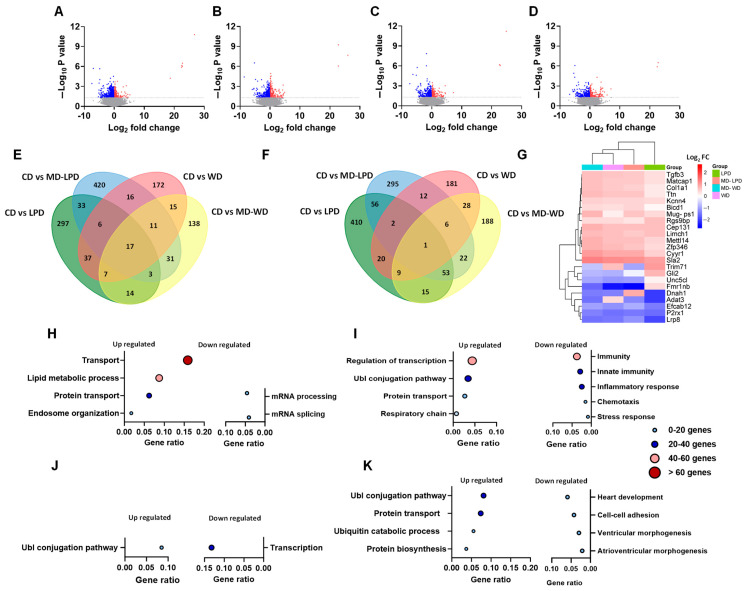
Comparison of maternal late gestation liver transcriptome in response to paternal diet. Volcano plot of differentially expressed hepatic genes between females mated to males fed either a control diet (CD) and (**A**) a low protein diet (LPD), (**B**) methyl donor-supplemented LPD (MD-LPD), (**C**) Western diet (WD) or, (**D**) methyl donor-supplemented WD (MD-WD). Significantly (*p* < 0.05) down- and up-regulated genes are shown in blue and red, respectively while non-differentially expressed genes are shown in grey. Venn diagrams showing the number of unique and common, significantly downregulated (**E**) and upregulated (**F**) genes in each group when compared to females mated with CD fed males. Heat map displaying all common genes (23 in total) in each group when compared to females mated with CD fed males (**G**). Gene ontology and pathway analysis of significantly up- and downregulated genes between females mated to males fed either a control diet (CD) and (**H**) LPD, (**I**) MD-LPD, (**J**) WD or, (**K**) or MD-WD. Colour and size of circles represent the number of genes differentially expressed in each pathway. Sequencing data are from 6 females per group, each mated with a separate male.

**Figure 4 nutrients-16-01879-f004:**
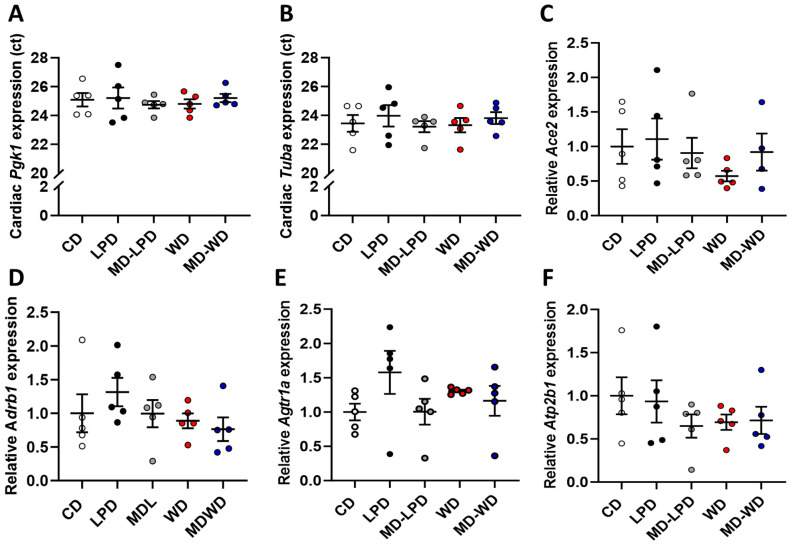
Analysis of maternal late gestation cardiac gene expression. The cycle threshold (Ct) values for the reference genes phosphoglycerate kinase 1 (Pgk1) (**A**) and tubulin, alpha 1a (Tuba1a) (**B**) in females mated to males fed either a control diet (CD), low protein diet (LPD), methyl donor-supplemented LPD (MD-LPD), Western diet (WD) or methyl donor-supplemented WD (MD-WD). Relative expression of angiotensin converting enzyme 2 (Ace2) (**C**), adrenergic receptor, beta 1 (Adrb1) (**D**), angiotensin II receptor, type 1a (Agtr1a) (**E**) and ATPase, Ca2+ transporting, plasma membrane 1 (Atp2b1) (**F**). *n* = 5 females per treatment group, each mated with a separate male. Data are mean ± SEM. Statistical differences were determined using a one-way ANOVA or Kruskal–Wallis test with post hoc correction.

**Figure 5 nutrients-16-01879-f005:**
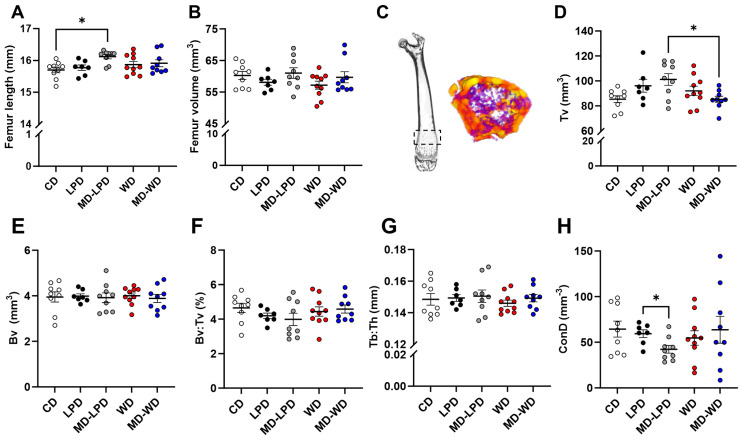
Assessment of maternal late gestation femur trabecular bone morphology. Whole femur length (**A**) and volume (**B**) in females mated to males fed either a control diet (CD), low protein diet (LPD), methyl donor-supplemented LPD (MD-LPD), Western diet (WD) or methyl donor-supplemented WD (MD-WD). Representative image of a whole scanned femur highlighting the trabecular bone region (dashed box) and the re-composed trabecular bone (**C**). Trabecular volume (Tv) (**D**), bone volume (Bv) (**E**), bone volume–trabecular volume ratio (Bv:Tv) (**F**), trabecular thickness (Tb:Th) (**G**) and connectivity (ConD) (**H**). *n* = 7–10 females per treatment group. Data are mean ± SEM. Statistical differences were determined using a one-way ANOVA or Kruskal–Wallis test with post hoc correction. * *p* < 0.05.

**Figure 6 nutrients-16-01879-f006:**
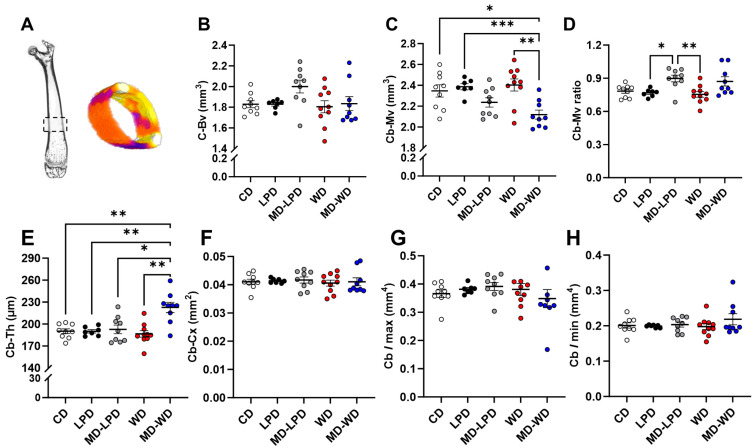
Assessment of maternal late gestation femur cortical bone morphology. Representative image of a whole scanned femur, highlighting the cortical bone region (dashed box) and the re-composed cortical bone (**A**). Cortical bone volume (C-Bv) (**B**), cortical bone medullary cavity volume (Cb-Mv) (**C**), cortical bone: medullary cavity ratio (**D**), cortical bone thickness (Cb-Th) (**E**), cortical bone cross section area (**F**), cortical bone maximum moment of inertia (Imax) (**G**) and minimum moment of inertia (Imin) (**H**) in females mated to males fed either a control diet (CD), low protein diet (LPD), methyl donor-supplemented LPD (MD-LPD), Western diet (WD) or methyl donor-supplemented WD (MD-WD). *n* = 7–10 females per treatment group. Data are mean ± SEM. Statistical differences were determined using a one-way ANOVA or Kruskal–Wallis test with post hoc correction. * *p* < 0.05, ** *p* < 0.01, *** *p* < 0.001.

## Data Availability

All data are either contained within the Appendix A or are available upon reasonable request sent to the corresponding author. Liver RNA-seq data have been submitted to the Gene Expression Omnibus (GEO) at NCBI under accession number: GSE265783 https://www.ncbi.nlm.nih.gov/geo/query/acc.cgi (accessed on 10 June 2024).

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
