# Peer review of "Sub-Optimal Paternal Diet at the Time of Mating Disrupts Maternal Adaptations to Pregnancy in the Late Gestation Mouse"

_nutrients, 2024, doi:10.3390/nu16121879_

Round 1

Reviewer 1 Report

Comments and Suggestions for Authors

This is an experimentally/analytically voluminous study on the potential effects of paternal diet on females' outcome during pregnancy. Per se, this is an important topic. Nevertheless, there are several major and minor issues to be addressed:

Major:

1. Title: Data do not justify the word 'perturbs'.

2. Abstract, l. 38F: For such statement, log-term follow-up data of dams and offsprings should have been shown. This holds, as many physiological data are not presented.

3. L. 97f : Folate and B-vitamins are not methyl donors but carriers. Vitamin B12 isn't mentioned as a variable.

4. L.105/ref 29: Why didn't the authors analyse seminal plasma (and further paternal parameters) as indicated in the reference. There is no single metabolic/biochemical data of the success of diet treatment of the males provided. Was mating and semination identically successful and rapid in animal groups? What was the effect of MDs?

L. 122 and supplementary Tab. 1:

5. The diets and groups are questionable in parts. A control group with added methyl donors is missing, precluding to address their basic effect.

6. Corn oil isn't the natural fat for mice, nor is it an adequate control diet for humans. Moreover, a diet free of cholesterol is vegan, but isn't traditional in most human societies (nor in mice except held experimentally!). Even in Western diet, vegetale and other fats make up a significant part of diet. Finally, the western diet group (WD) has no methionine, and therefore isn't comparable with the other groups. It is not declared whether the variables are what's added or what's in the diets. Please specify B-vitamins.

7. 2.2, l. 141ff: Data on paternal metabolites and success of 'methyl donation' is missing. Importantly, there is no quantification of choline, betaine, methionine and any of its derivatives/downstrem metabolites. Lipid analysis in serum/plasma  of the dams is missing. Histology of the liver (triglyceride content) is missing.

8. 2.7, l. 264ff: Mention the mode of data presentation (means +/- standard deviations or medians and interquartile ranges, both here and in the figures. Means +/- SEM is misleading as it depends on group size. May be, it's more useful to present data uniformly as medians and IQR. Were data corrected for both multiple group comparisons and repeated-measurement ? Define the numbers of animals per group here.

9. Figures: The fonts within the figures are far too small. They mustn't be much smaller than those of the text, particularly with such voluminous figures! The significance symbols should be easily visible in figures.

10. Some figures may be better shown as a table, like Fig. 1 K to O as well as Fig 4, 5 and 6 B to H.

Discussion, l. 539:

11. The discussion is too long. Everything should be written more focussed.

12. Given the small number of experiments, missing MD-CD group, and coverage of so many parameters, the authors should state that it is an explorative study. This should made clear already in the introduction.

Minor:

1. Title: double space between 'to' and 'pregnancy'.

2. L. 56: 'associates'.

3. L.59ff: To justify the used model, a statement should be made that this is identically or similarly true for both the human being and mouse.

4. L. 101: Replace 'conception' by 'procreation/siring'.

5. L. 122: replace sugar by carbohydrate and mention the different proportions of starch versus sucrose.

6. L. 135: grammar and spaces: at '80° C'

7. L. 153 and l. 163: specify teh analysed tissue.

8. 2.6, l. 248ff: The authods should have measured physiological data of cardiac function.

9. 2.7: 'Shapiro' rather than 'Shaprio'.

10. 3.1, l. 272-275: This is introduction (1st sentence) and methods (2nd sentence).

11. L. 620: replace 'and' by 'of'.

Author Response

Pease see the attachment.

Reviewer 2 Report

Comments and Suggestions for Authors

Dear Editor,

I carefully read the manuscript "Sub-optimal paternal diet at the time of mating perturbs maternal adaptations to pregnancy in the late gestation mouse".

This is an interesting study, with statistical analysis adequately described and properly  performed.

My comments and suggestions for the authors are the following:

 - The limitations of the study should be further and more deeply discussed by the authors

 - Some of the references need to be updated (for example, ref. 1 is outdated, as well as ref. 7 and ref. 17 and 18). In particular, ref. 18 should be read in light of the most recent position of the European Atherosclerosis Society, for example.

Reviewer 3 Report

Comments and Suggestions for Authors

This is a good work, well designed and presented in comprehensive language and descriptive plots. Yet, the samples are very small and the results may be considered fallacious due to increased bias.  Small samples fallacy is confronted either by using non-parametric tests or even better convert the present results into Hedge's g effect size and 95%CI.

This is crucial for precision and accuracy and validity reasons.

Finally, authors should write clearly the limitations of their study.
